# Effect of Pre-Soaking Treatment Method of Plant-Based Aggregate on the Properties of Lightweight Concrete—Preliminary Study



Ming Kun Yew [1,*], Ming Chian Yew [2], Jing Han Beh [3], Foo Wei Lee [1], Siong Kang Lim [1], Yee Ling Lee [1], Jee Hock Lim [1] and K. I. Syed Ahmed Kabeer [4]

1 Department of Civil Engineering, Lee Kong Chian Faculty of Engineering and Science, Universiti Tunku Abdul Rahman, Cheras, Kajang 43000, Malaysia; leefw@utar.edu.my (F.W.L.); sklim@utar.edu.my (S.K.L.); yllee@utar.edu.my (Y.L.L.); limjh@utar.edu.my (J.H.L.)
2 Department of Mechanical and Material Engineering, Lee Kong Chian Faculty of Engineering and Science, Universiti Tunku Abdul Rahman, Cheras, Kajang 43000, Malaysia; yewmc@utar.edu.my
3 Department of Architecture and Sustainable Design, Lee Kong Chian Faculty of Engineering and Science, Universiti Tunku Abdul Rahman, Cheras, Kajang 43000, Malaysia; behjh@utar.edu.my
4 School of Architecture and Interior Design, Faculty of Engineering and Technology, SRM Institute of Science and Technology, Kattankulathur, Chennai 603203, India; syedk@srmist.edu.in
* Correspondence: yewmk@utar.edu.my

**Abstract:** This research investigates the effect of pre-soaking treatment on plant-based aggregate using a wet grout binder to formulate a high-strength lightweight concrete (HSLWC). Surface modification utilising a novel grout soaking technique with various water-to-cement (w/c) ratios has indicated a new method of approach for the recent development of lightweight plant-based aggregate (LWPA). In this experiment, the fresh and hardened properties of modified LWPA lightweight concrete were assessed by verifying their workability, densities, compressive and split tensile strengths towards the modulus of elasticity. The results showed that pre-soaking plant-based lightweight aggregate (w/c: 0.6, 0.8, 1.0 and 1.2) slightly increased the density of the samples compared to untreated LWPA. The oven-dry density of treated and untreated LWPA is controlled in the range of HSLWC. The outcomes indicated that the workability of the surface-modified LWPA is significantly improved: up to 40% in 6 min for the (TDS)/0.6 sample compared to the original LWPA. The mechanical properties of the LWPA concrete with the surface modification method exhibit a substantial increment of compressive strength, split tensile strength and the modulus of elasticity; recorded at 22%, 26% and 34%, respectively. Significantly, the findings from this experiment reveal that the pre-soaking treatment method on LWPA is shown to be a highly recommended technique in improving interfacial bonding while maintaining its performance as one of the most promising solutions to improve the properties of lightweight concrete.

**Keywords:** density; pre-soaking; high strength concrete; lightweight plant-based aggregate; mechanical properties; environmentally friendly





## 1. Introduction

Lightweight concrete is the most popular building material for constructing certain civil infrastructure projects. In general, more than ten billion tons of concrete is produced annually, consisting of natural aggregate such as crushed rock, fine sand, and gravel [1,2]. The construction sector contributes substantially to society and economic development by improving well-being and quality of life [3–5]. The construction industry also significantly impacts the environment, accounting for a considerable portion of natural resource depletion. The current trend in the construction industry is towards using alternative renewable construction materials. Therefore, growing awareness has been paid to the various surface

modification methods of renewable aggregate to minimise the impact towards resources, thereby reducing environmental impact to produce environmentally friendly concrete [6–9].

The utilisation of lightweight crushed oil palm shell (OPS) as a substitute for conventional coarse aggregates has been used since early 1984 in Malaysia by Abang [10]. The use of lightweight plant-based aggregate in concrete can provide possible solutions to mitigate the depletion problem of natural resources. Recently, many researchers have utilised renewable and recycled lightweight plant-based aggregate (LWPA) such as wood, palm kernel shells, peach shells, mussel shells, coconut shells, bamboo and apricot shells to produce lightweight concrete [11–15]. The advances of incorporating plant-based aggregate in lightweight concrete serve to further reduce the concrete's self-weight, and one way to alleviate the damage inflicted on the natural environment is by using sustainable building materials [16,17]. Therefore, the utilization of unwanted waste from agricultural activities such as OPS in concrete can provide sustainable development of construction by contributing to reducing greenhouse gas emissions. Many scientists have found that the utilization of effective surface modification methods on plant-based aggregates can produce better concrete quality [18–23]. The modification techniques consist of carbonization, particle shaping, microwave heating and soaking of chemical solutions to enhance bonding. By employing these strategies, the adhered shell can be strengthened and made more durable, ensuring its longevity and effectiveness.

The Malaysian Palm Oil Council (MPOC) reports that Malaysia is one of the largest producers of crude palm oil and is expected to export approximately 0.19 billion tons of crude palm oil each year, which is 12% of global palm oil [24]. Indonesia and Malaysia are responsible for supplying 34% of the global vegetable oil demand [25]. The oil palm fruit in Malaysia can be categorized into two main types: tenera and dura [26]. The total export of oil palm products has reached more than RM 67.5 million and contributes 37.7% of Malaysia's agricultural GDP. It has been reported that 2.7 million hectares of land area is covered with oil palm plantations and more new oil palm plantations are being introduced and developed [27]. The production of crude palm oil (CPO) is estimated to reach 5.5 million and increases every year. With the demand of palm oil, a large amount of oil palm by-product waste is produced. From recent studies, many scientists have investigated oil palm shells as a bio-based lightweight aggregate to produce green lightweight concrete [28–30]. Lightweight concrete can be produced by using a variety of materials, such as lightweight, fine, coarse and bubble foamed aggregate [31–33]. Some of the lightweight aggregates used are perlites, pumice and scoria. Furthermore, lightweight foamed concrete is produced by adding a foaming agent in which air-voids can be entrapped in the mortar mix at up to 75%. According to Loh et al., one of the most popular techniques for producing lightweight concrete is incorporating lightweight aggregate [34]. From the previous study, desirable characteristics for sustainable building materials in terms of high strength lightweight concrete with incorporated plant-based aggregate are those that fall within the range of 40–54 MPa of compressive strength and less than 1900 kg/m$^3$ of oven-dry density [26].

Most current research on OPS lightweight concrete focused on the investigation of its surface modification with the heat-treatment and grout spray coating methods [7,22]. However, no information is available regarding pre-soaking treatment techniques on plant-based aggregates. Therefore, implementation of innovative techniques on plant-based aggregate with the consideration of effective methods to mitigate environmental issues are strongly recommended. Thus, the influence of pre-soaking techniques with various w/c formulation ratios (0.6, 0.8, 1.0 and 1.2) on dura and tenera plant-based aggregates in terms of mechanical and durability properties will be investigated.

## 2. Materials and Methods

### 2.1. Materials

Locally produced Type I 43 grade Ordinary Portland Cement conforming to the Malaysia Standard was chosen, and 5% of silica fume containing densified class pozzolana was adopted to be the supplementary cementitious material. The chemical and physical

compounds of the cement are provided in Table 1. The average particle size of OPC and SF are 25 µm and 15 µm, with a specific gravity of 3510 cm$^2$/g and 2.10 g/cm$^3$, respectively. In this study, portable water and polycarboxylic ether superplasticiser were used to prepare all the concrete mixtures. Natural river sand and crushed dura shell (DS) and tenera shell (TS) particles having an average size of 9.5 mm were utilized. Shafigh et al. [35] found that a 9.5 mm maximum size of OPS aggregate showed an increment of 6% compressive strength compared to 12.5 mm. Therefore, 9.5 mm maximum size of DS and TS is recommended, as shown in Figure 1. The grout soaking techniques with different w/c formulations (0.6, 0.8, 1.0 and 1.2) will be applied on the DS and TS surface until the coats are evenly distributed. It is important to note that different types of plant-based aggregates may require different soaking times and temperatures. Therefore, plan-based aggregates after a pre-soaking treatment with heating temperature up to 75 °C and a time interval of 0.5 h will be investigated. After 24 h, all the dried DS and TS will be used as coarse aggregates for mixtures, as shown in Figures 2 and 3. The physical properties of river sand and coated DS and TS aggregates are shown in Table 2. Furthermore, the average grading of coated and uncoated DS and TS aggregates are as illustrated in Table 3. In addition, the mixing method of untreated and treated LWPA concrete was performed, as shown in Figure 4.

**Table 1.** Chemical and physical properties of Ordinary Portland Cement.

| Chemical Analysis | (%) | Physical Properties | Unit | |
|---|---|---|---|---|
| SiO$_2$ | 21.5 | Specify gravity | g/cm$^3$ | 3.14 |
| Al$_2$O$_2$ | 5.9 | Specific surface area | cm$^2$/g | 3510 |
| Fe$_2$O$_2$ | 3.4 | Compressive strength | MPa | |
| CaO | 59.8 | 3 days | - | 23.3 |
| SO$_3$ | 4.3 | 28 days | - | 46.2 |
| MgO | 2.9 | Flexural strength | MPa | |
| Loss on ignition | 0.6 | 3 days | - | 4.2 |
| | | 28 days | - | 7.3 |
| | | Initial setting time | min | 230 |

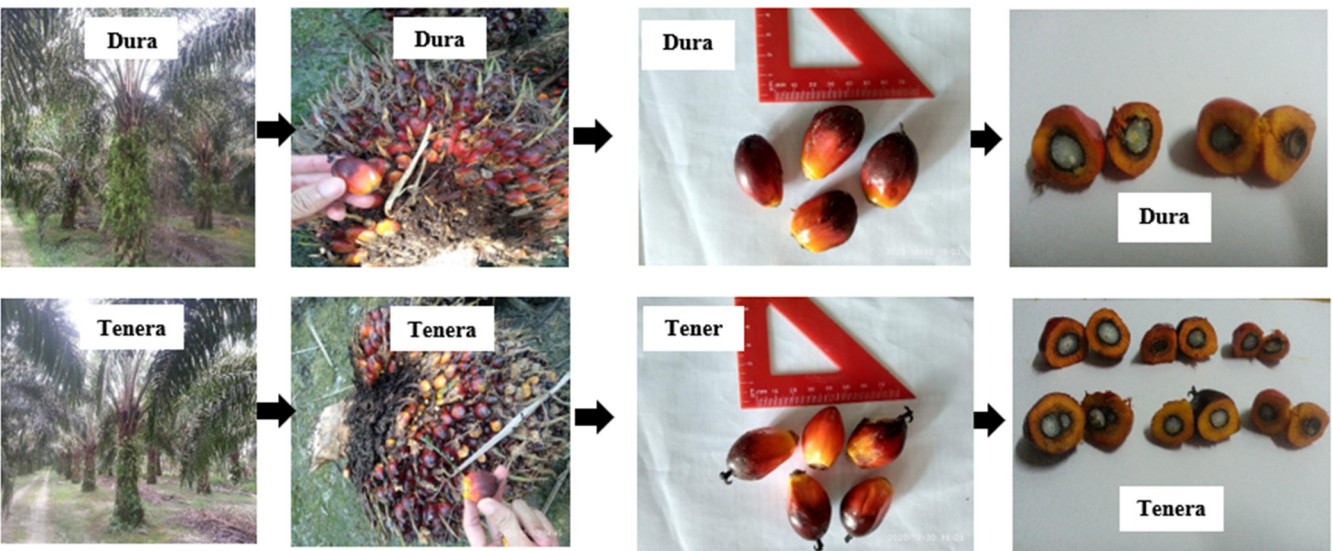

**Figure 1.** Dura and tenera oil palm fresh fruit bunches taken from 5–10-year-old oil palm trees.

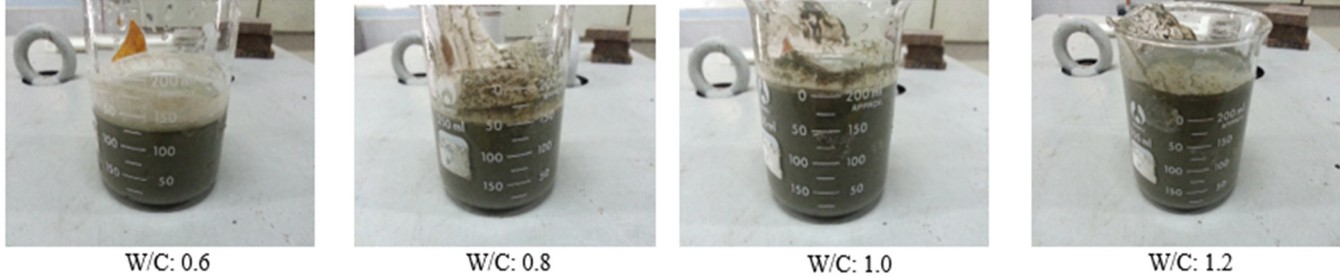

**Figure 2.** Different water/cement (w/c) ratio from 0.6–1.2.

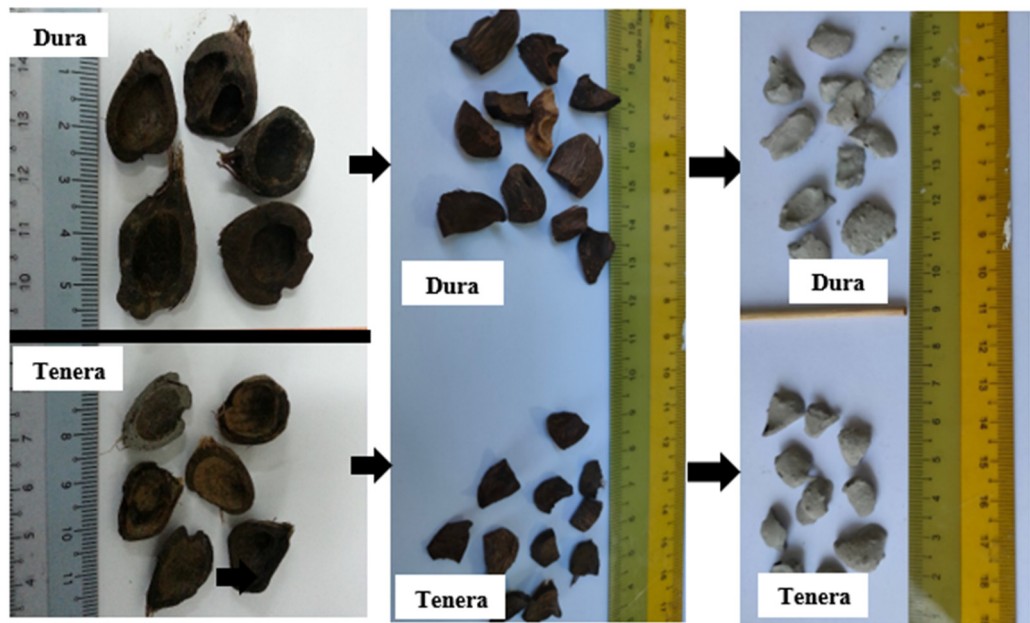

**Figure 3.** Pre-treated grout soaking on dura and tenera oil palm shell.

**Table 2.** The properties of sand, untreated and treated LWPB aggregate.

| Physical Property | Unit | Fine Aggregate (River Sand) | Coarse Aggregate | | | | | | | | | |
|---|---|---|---|---|---|---|---|---|---|---|---|---|
| | | | DS/0 | TDS/0.6 | TDS/0.8 | TDS/1.0 | TDS/1.2 | TS/0 | TTS/0.6 | TTS/0.8 | TTS/1.0 | TTS/1.2 |
| Specific gravity | g/cm² | 2.67 | 1.23 | 1.47 | 1.45 | 1.40 | 1.38 | 1.12 | 1.33 | 1.31 | 1.28 | 1.26 |
| Bulk density | kg/m² | 1568 | 623 | 656 | 654 | 647 | 646 | 618 | 641 | 647 | 637 | 635 |
| Fineness modulus | | 2.71 | 5.0 | 5.7 | 5.5 | 5.4 | 5.3 | 2.4 | 5.2 | 5.1 | 4.6 | 4.2 |
| Water absorption (20 min) | % | - | 7.0 | 3.52 | 3.55 | 3.68 | 3.70 | 6.8 | 3.37 | 3.35 | 3.45 | 3.40 |
| Water absorption (24 h) | % | 1.2 | 17.0 | 14.6 | 14.9 | 15.2 | 15.4 | 15.2 | 12.3 | 12.5 | 13.0 | 13.5 |
| Aggregate impact value | % | - | 2.38 | 3.56 | 3.52 | 3.50 | 3.48 | 2.27 | 3.55 | 3.53 | 3.50 | 3.40 |
| LA abrasion value | % | - | 7 | 18 | 16 | 13 | 12 | 5 | 14 | 12 | 10 | 8 |
| Flakiness index | % | - | 35 | 21 | 23 | 27 | 29 | 31 | 22 | 24 | 26 | 28 |
| Surface texture | - | Rough | Rough | Rough | Rough | Rough | Rough | Rough | Rough | Rough | Rough | Rough |

**Table 3.** Grading of treated and non-treated OPS aggregates.

| Sieve Size (mm) | Cumulative % by Weight Passing Sieve Size | | | |
|---|---|---|---|---|
| | DS (12.5 mm) | TS (12.5 mm) | TDS (9.5 mm) | TTS (9.5 mm) |
| 20 | 100 | 100 | 100 | 100 |
| 12.5 | 100 | 100 | 100 | 100 |
| 9.5 | 84.15 | 84.50 | 100 | 100 |
| 8 | 59.60 | 59.90 | 93.35 | 94.75 |
| 4.75 | 24.50 | 22.70 | 26.20 | 25.80 |
| 2.36 | 3.50 | 4.60 | 4.85 | 4.90 |

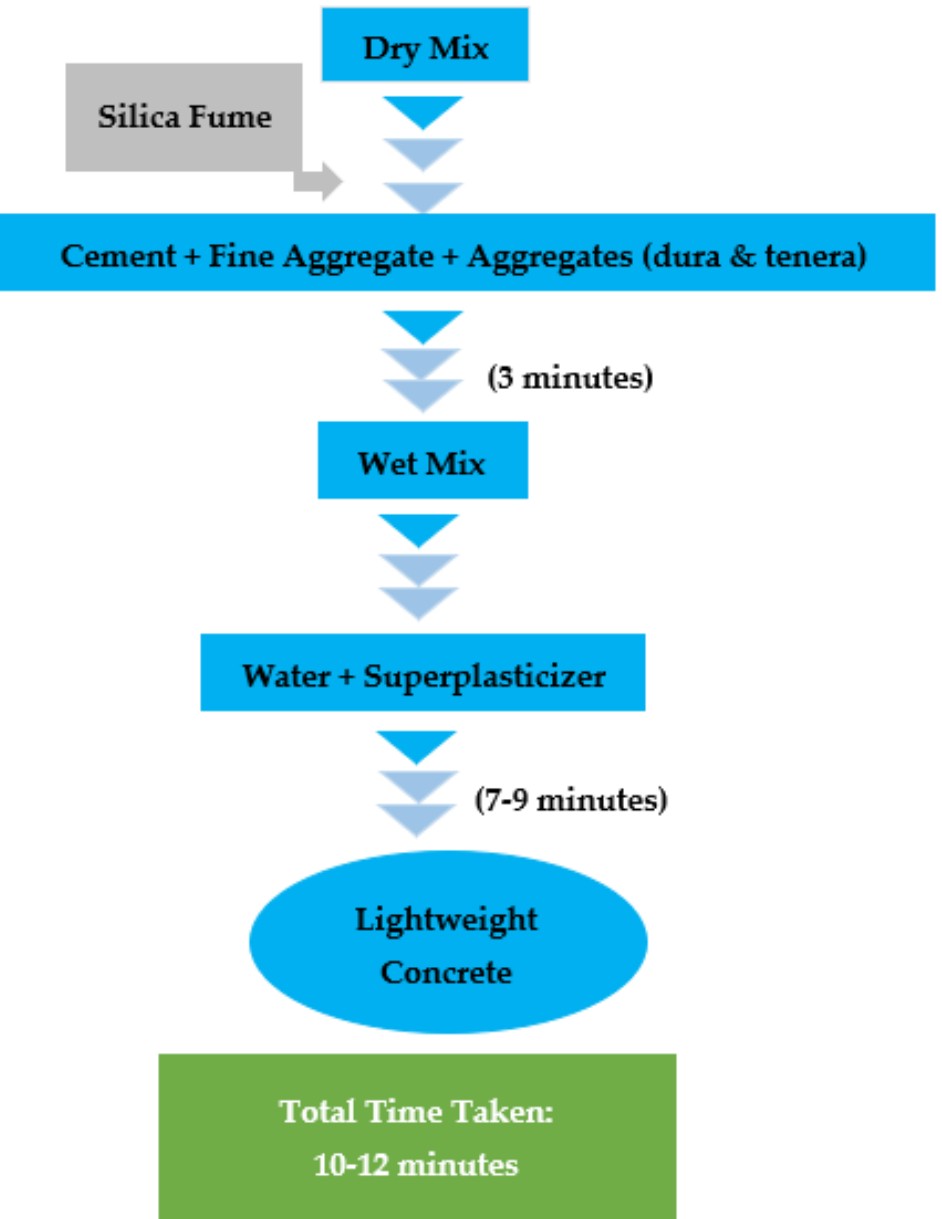

**Figure 4.** Dry mixing method of untreated and treated LWPA concrete.

### 2.2. Concrete Sample Preparation and Methods of Testing

A detailed proportioned mix of untreated and treated LWPA concrete was proposed, as illustrated in Table 4. The fresh concrete was put in steel moulds, which were properly lubricated before placement. The 100 mm × 100 mm × 100 mm cube, 100 mm × 200 mm cylinder and 150 mm × 300 mm cylindrical specimens were used to find the compressive strength, splitting tensile strength and Young's modulus of elasticity, and all the concrete specimens were kept in a water tank for a duration of 1, 3, 7, 28 and 56 days for water curing. The fresh properties of concrete were examined immediately after mixing. In addition, different tests were performed according to compressive strength: BS EN 12390-3 [36], splitting tensile strength: BS EN 12390-6 [37] and modulus of elasticity: BS EN 12390-13 [38], respectively by using a compression machine with a capacity of up to 3000 kN. Furthermore, water absorption was examined for all the specimens at 28 days.

**Table 4.** The mix proportion of untreated and treated LWPA concrete (kg/m$^3$).

| Mix Code | Cement | Silica Fume | Water | W/C | Sand | SP | OPS | Temperature (°C) | Heating Time (h) | Grout Coating (W/C) |
|----------|--------|-------------|-------|-----|------|-----|-----|------------------|------------------|---------------------|
| DS/0 | 490 | 25 | 154 | 0.3 | 908 | 6.2 | 355 | Room temperature | 0 | 0 |
| TDS/0.6 | 490 | 25 | 154 | 0.3 | 908 | 6.2 | 425 | 75 | 0.5 | 0.6 |
| TDS/0.8 | 490 | 25 | 154 | 0.3 | 908 | 6.2 | 410 | 75 | 0.5 | 0.8 |
| TDS/1.0 | 490 | 25 | 154 | 0.3 | 908 | 6.2 | 395 | 75 | 0.5 | 1.0 |
| TDS/1.2 | 490 | 25 | 154 | 0.3 | 908 | 6.2 | 380 | 75 | 0.5 | 1.2 |
| TS/0 | 490 | 25 | 154 | 0.3 | 908 | 6.2 | 345 | Room temperature | 0 | 0 |
| TTS/0.6 | 490 | 25 | 154 | 0.3 | 908 | 6.2 | 420 | 75 | 0.5 | 0.6 |
| TTS/0.8 | 490 | 25 | 154 | 0.3 | 908 | 6.2 | 405 | 75 | 0.5 | 0.8 |
| TTS/1.0 | 490 | 25 | 154 | 0.3 | 908 | 6.2 | 390 | 75 | 0.5 | 1.0 |
| TTS/1.2 | 490 | 25 | 154 | 0.3 | 908 | 6.2 | 375 | 75 | 0.5 | 1.2 |

## 3. Results and Discussion

### 3.1. Fresh Properties and Densities

In this research, the amount of slump is measured and compared to the requirements for a particular application to ensure that the concrete has the right consistency to be placed and compacted properly. The influence of treated and untreated LWPA was assessed. In Figure 5, the age category of the fresh concrete specimens that can affect the slump value are presented. The slump value loss of the mixture for all samples is a widely used method to measure the consistency and workability of fresh concrete at 3, 6 and 9 min. The highest slump value of 155 mm was obtained for the TDS/0.6 mix, the minimum slump value of 90 mm was obtained for the TS/0 mix. For the mixes, good workability requires a longer time for the reaction between the cement and superplasticiser. It can be noted that the improvement of workability due to the lower pre-soaking w/c ratio is denser compared to the untreated shell. On other hand, the cohesion and bond strength between the pre-soaking shells that can enhance the workability of fresh concrete is expected to reduce friction compared to the untreated shell. The use of treated peach shell as a lightweight aggregate in concrete has shown promising results in reducing water absorption between 8% to 15% compared to the control specimen. This is due to the fact that the treatment process removes the organic matter and reduces the porosity of the shells, resulting in a denser and more durable aggregate [39].

### 3.2. Densities (Demoulded, Air-Dry and Oven-Dry)

Lightweight plant-based aggregate (LWPA) concrete can be categorized as a special type of concrete with a characteristic oven-dried density not more than 2000 kg/m$^3$ [40] Among the three types of density (DD: demoulded density, ADD: air-dry density, and ODD: oven-dry density), all specimens fell within the range of the lightweight concrete category, as presented in Figure 6. The oven-dry density and air-dry density of untreated shell mixes ranged between 1897–1908 kg/m$^3$ and 1973–1998 kg/m$^3$, respectively. From the DD results, it can be observed that 5 mixes (TTS/0.8, TTS/0.6, TDS/1.0, TDS/0.8 and TDS/0.6) slightly exceeded 2000 kg/m$^3$. All mixes fulfilled the lightweight concrete requirements in accordance with the oven-dry density standard. From the results, the replacement of the untreated DS and TS with various ratios of the pre-soaking method on shell aggregates (TDS/0.6–1.2 and TTS/0.6–1.2) marginally increased the DD, ADD and ODD at about 7%, 10% and 12%, respectively. The increment in the density of lightweight concrete is due to a higher specify gravity with pre-soaking shell aggregates, especially for TDS/0.6 and/TTS/0.6. According to Mannan et al. [21], it has been reported that treatment with polyvinyl alcohol also slightly enhanced the overall density of bio-based lightweight concrete.

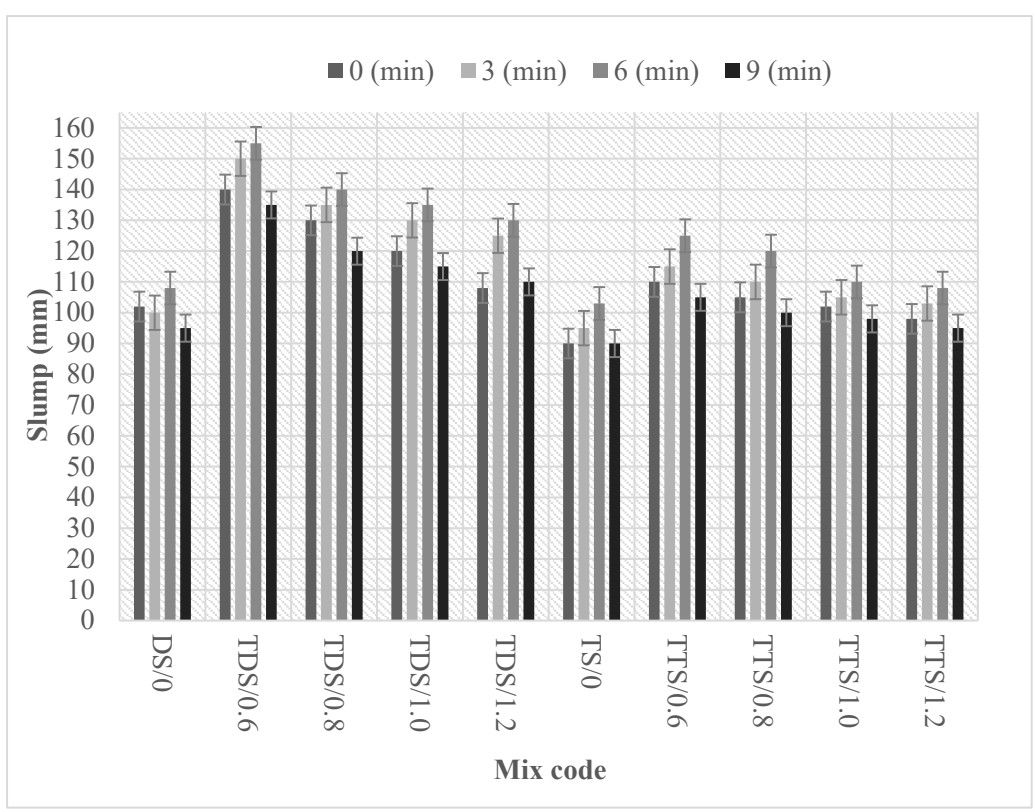

**Figure 5.** Effect of untreated and treated DS and TS versus slump values.

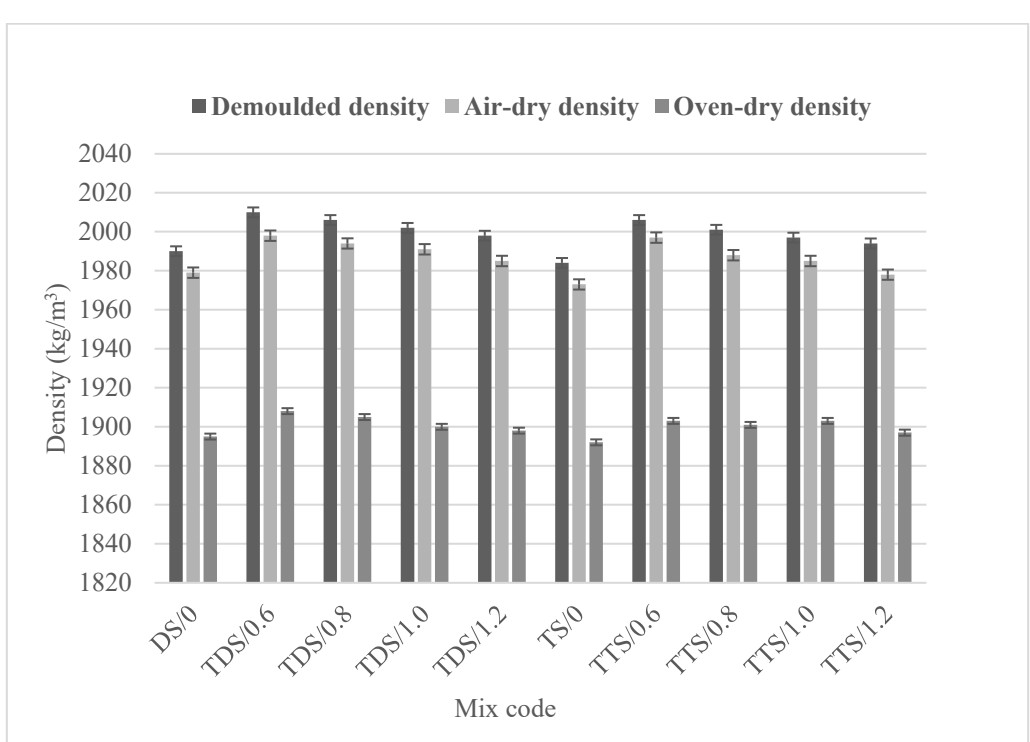

**Figure 6.** Untreated and treated DS and TS versus various densities.

*3.3. Mechanical Properties of Concrete*

Compressive Strength Test

The compressive strength results of all the specimens with the pre-soaking modification method on LWPA concretes at the age of 1, 3, 7, 14, 28 and 56 days are presented

in Figure 7. The mix with the pre-soaking treated shells increased the strength versus the control specimens. It can be noted that the TDS/0.6 concrete mix achieved the highest compressive strength of about 57 MPa at 28 days. The specimens with the pre-soaking treatment method obviously improved—from TS/0 to DS/0, TTS/0.6–1.2 and TDS/0.5–1.2—at all ages. In all the specimens from TS/0 to TTS/0.6–1.2, compressive strength was enhanced by 7.7–15.4% at the 1st, 3.8–15.2% at the 3rd, 5.3–11.3% at the 7th, 6.5–11.3% at the 14th, 4.0–17.1% at the 28th and 4.2–17.2% at the 56th day, respectively. The cube compressive strength increased significantly; by 22.2% for the TDS/0.6 mix when compared to the control mix at 28 days. The thickness and toughness of LWPA shells corresponding to the pre-soaking treatment method with various ratios of w/c increases significantly for TDS/0.6. According to Ryu et al. [40], the improvement of compressive strength is due to the interfacial transition zone (ITZ), which is subjected to the cohesive bonding strength between the cement paste and LWA. Furthermore, the compressive strength enhancement of the concrete cube is more prominent at the latter stages due to the special impact of the pre-soaking treatment method. The surface cracks of the untreated and treated concrete cube are shown in Figure 8. It was noted that the pre-soaking treatment method on DS and TS aggregate aided in filling the voids and reducing the cracks of the lightweight concrete. From Figure 9, the pre-soaking treated shells aggregate-cement paste showed improvement in the interfacial area and showed successive reduction in the creation of micro-cracks as compared to the untreated shell lightweight concrete. In addition, the SEM images of shell aggregate before and after pre-soaking treatment are shown in Figure 10. The contribution of the 'soaking' effect that helped to improve the ITZ of lightweight concrete during the strengthening process should be noted. Martirena et al. [41] also reported similar positive effects provided by surface coating for recycled aggregates.

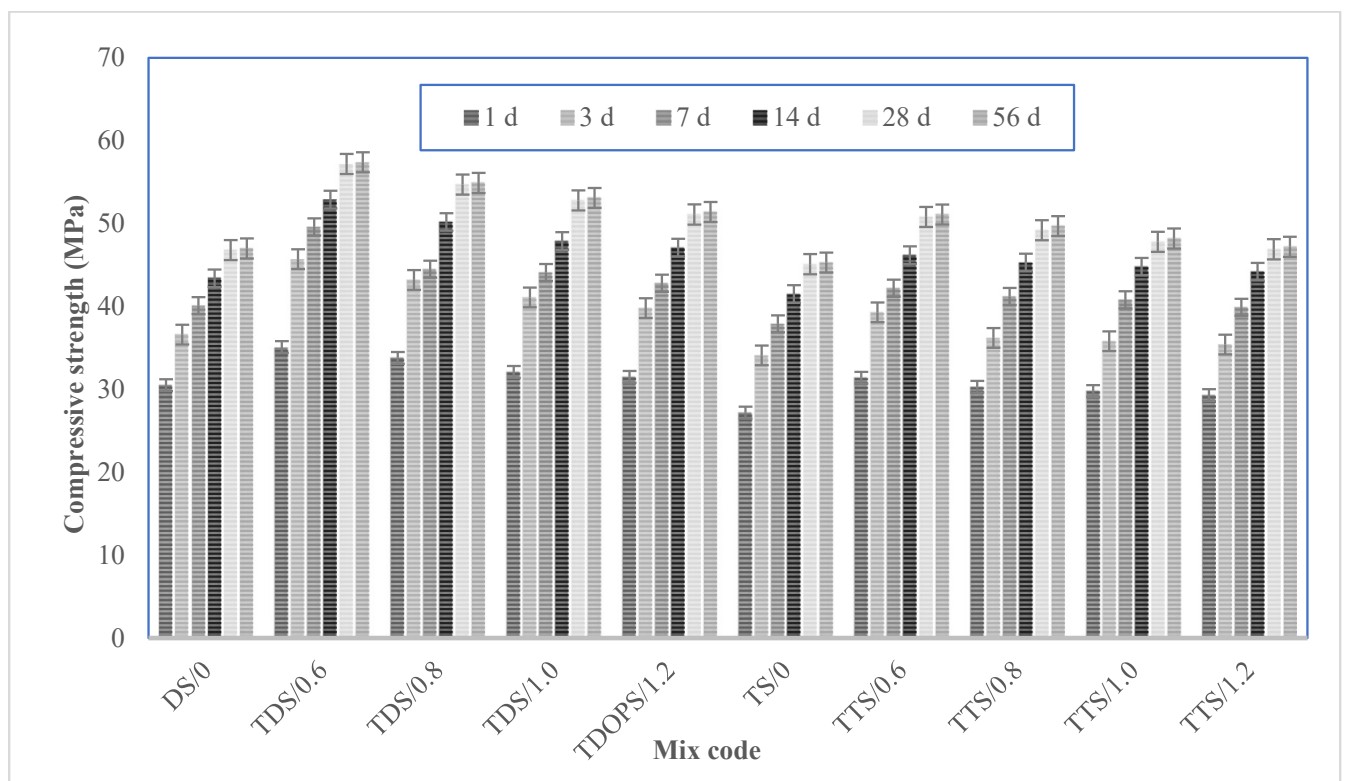

**Figure 7.** Compressive strength of untreated and treated LWPA concrete under moist curing.

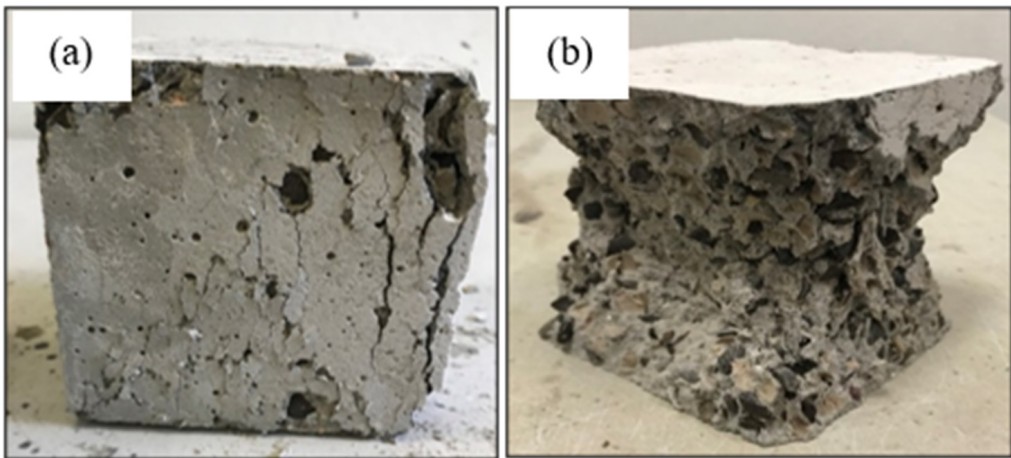

**Figure 8.** Treated (**a**) and untreated (**b**) crack surfaces of OPS aggregate for cube compressive strength test.

### 3.4. Split Tensile Strength Test

The split tensile strength test results of all the (untreated and treated LWPA) concrete mixers are illustrated in Figure 11. The incorporation of the pre-soaking treatment method aggregates reached a greater tensile strength compared to the control mix. The splitting tensile strength of TDS/0.6–1.2 prepared with treated dura shell at 28 days was extensively strengthened (by 9.5–26.0%) as compared to non-treated LWPA concrete. The splitting tensile strength of TTS/0.6–1.2 with pre-treated grout soaking improved simultaneously when compared to untreated LWPA concrete at 7 days and 28 days. The TDS and TTS modifications with different w/c formulations from 0.6 to 1.2 raise a new developed binding property between the shell aggregate and mortar in LWPA concrete.

The linear correlations between the compressive strength and the split tensile strength test of pre-treated grout soaking with different w/c (0.6–1.2) formulations of dura and tenera shells at 28 days are presented in Figure 12. The fitting correlation coefficient ($R^2$) for both treated dura and tenera reached up to 0.98 and 0.99, respectively. It indicated a consistent relationship between the compressive strength and the split tensile strength. Through the fitted equations, the splitting tensile strength of the treated LWPA concrete can be well predicted by its compressive strength. The following two equations are proposed for different types of treated shells (dura and tenera) to connect the $F_t$ and the cube strength of LWPA concrete.

$$F_{t1} = 1.47 f_{cu}^{\frac{1}{2}} - 6.60 \tag{1}$$

$$F_{t2} = 1.65 f_{cu}^{\frac{1}{2}} - 7.76 \tag{2}$$

where $F_{t1}$ and $F_{t2}$ and $f_{cu}$ are the splitting tensile (MPa) of treated dura and tenera as well as cube strengths in MPa, respectively.

### 3.5. Strength Relationship

The correlation between the early age strength at 1, 3, and 7 days, and the 28-day strength for untreated and treated lightweight plant-based concrete is significant, as shown in Figure 13. It can be noted that there is an appropriate linear growth for mixtures of untreated and treated LWPA concrete between the early stage (1 day, 3 days and 7 days) and those at the 28-day compressive strength stage. The graph shows a highly correlated coefficient with an $R^2$ value within the range of 0.93–0.96. According to Frost, it a trend line curve with a regression above 0.8 is classified as exceptional [42]. Equations (3)–(5) are recommended to assess the compressive strength of the cube at the early stage (1 day, 3 day and 7 day) strength values.

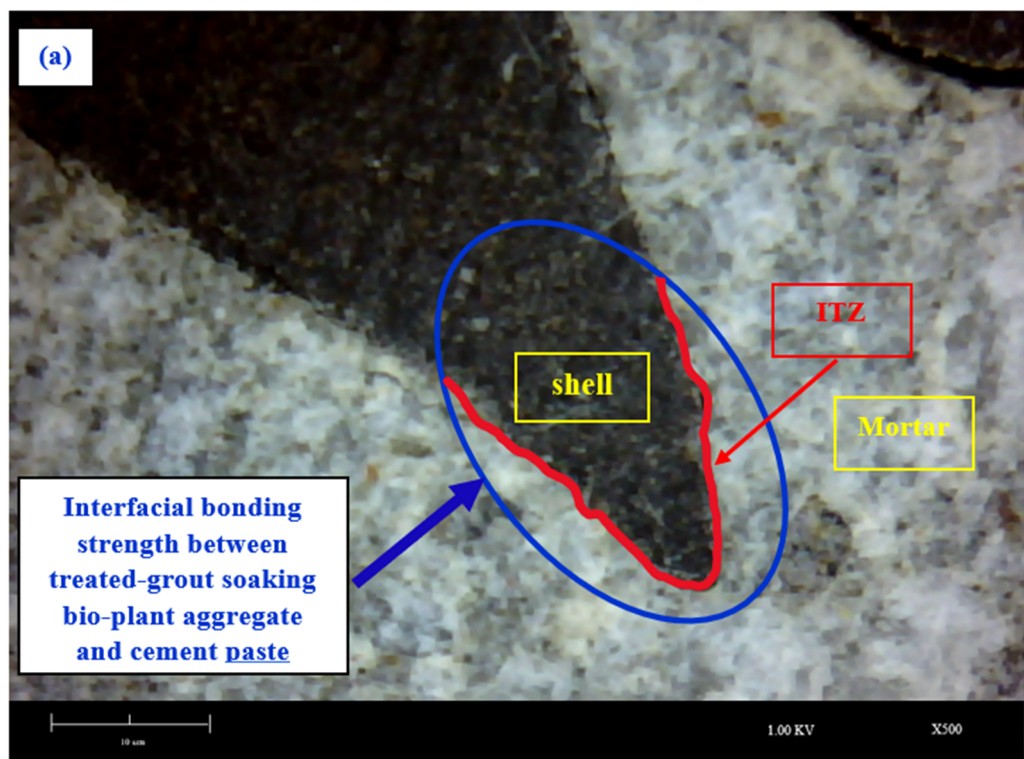

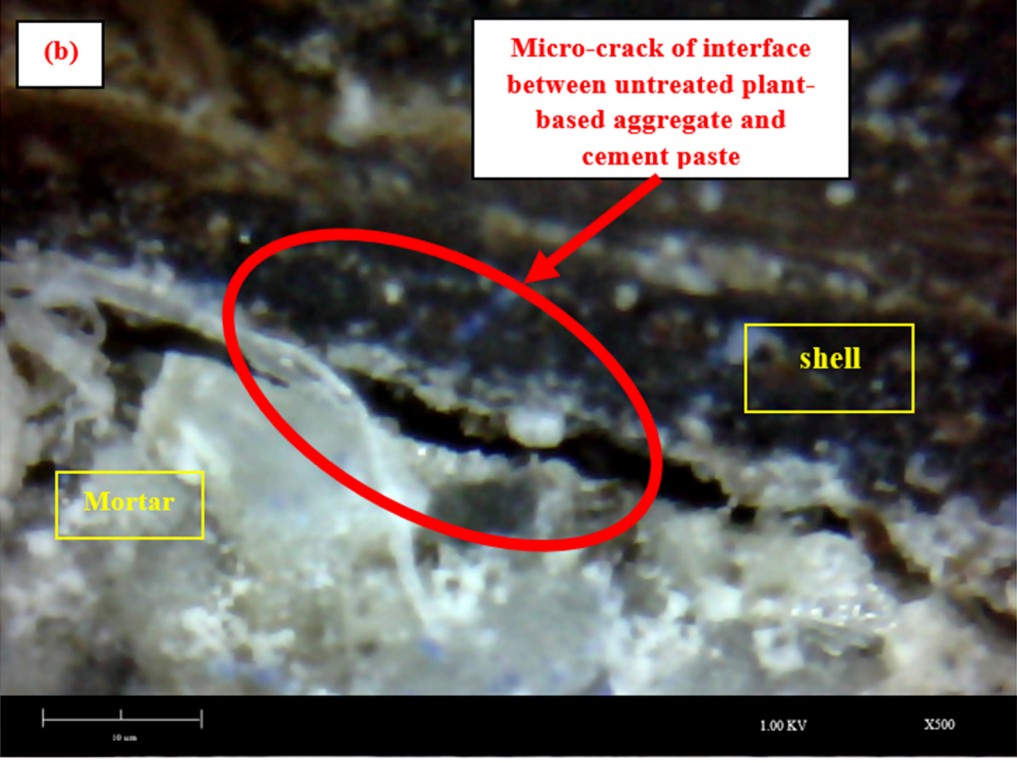

**Figure 9.** (**a**) Pre-treated grout soaking and (**b**) untreated interfacial bonding between the cement paste and shell aggregate.

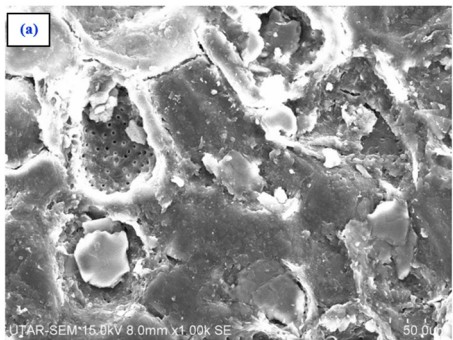
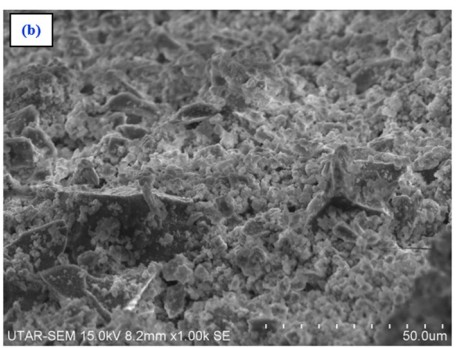

**Figure 10.** (**a**) The SEM pictures of aggregate (**a**) before treatment and (**b**) after treatment.

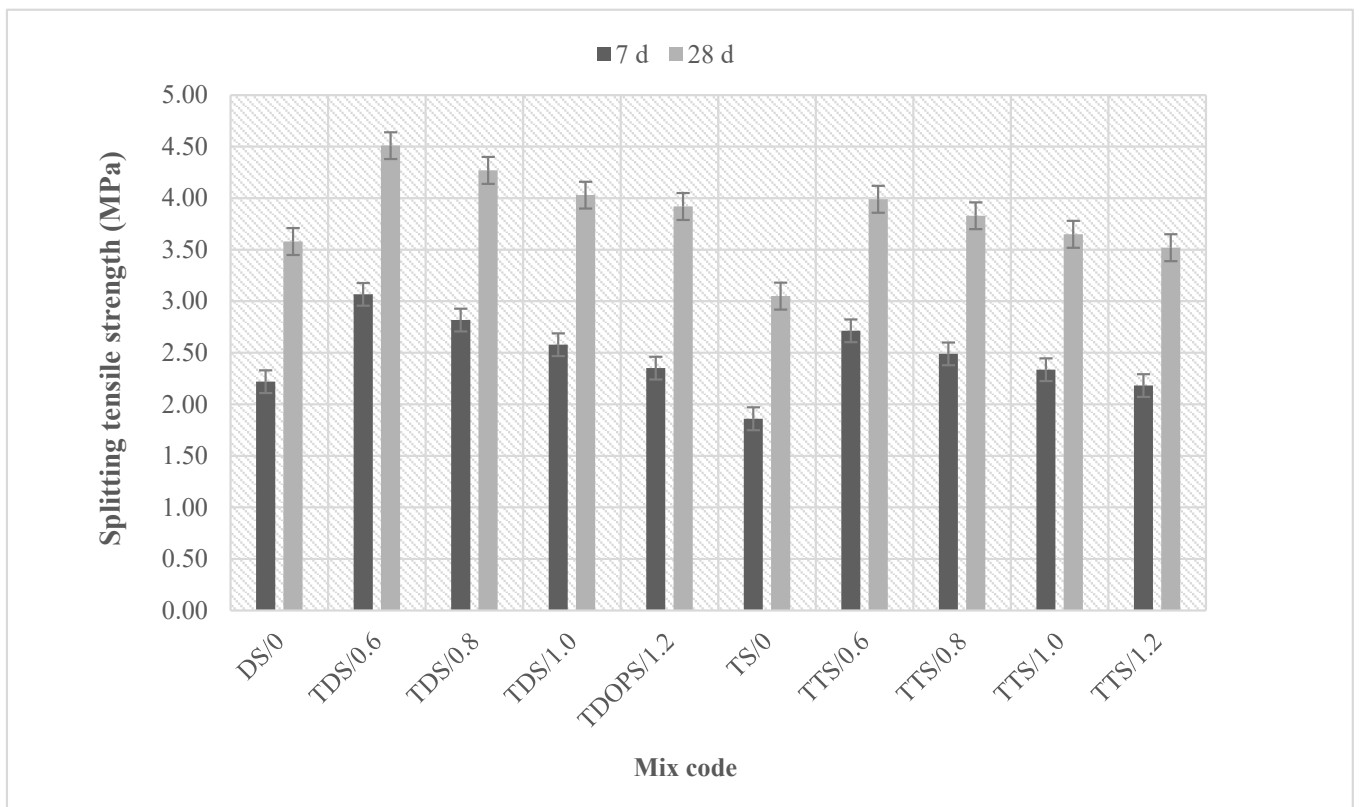

**Figure 11.** The splitting tensile strength of untreated and treated LWPA concrete.

$$F1 = 1.244 \ (E_1) + 42.16 \qquad (3)$$

$$F3 = 1.346 \ (E_3) + 25.99 \qquad (4)$$

$$F7 = 1.624 \ (E_7) + 9.82 \qquad (5)$$

where, *F1*, *F3* and *F7* represent the cube compressive strength (MPa), and $E_1$, $E_3$ and $E_7$ represent the early stage at 1 day, 3 days and 7 days of compressive strength, respectively.

According to Yew et al. [22], the high relationship coefficient was observed in oil palm shell (OPS) concrete made with heat-treated and crushed OPS aggregates.

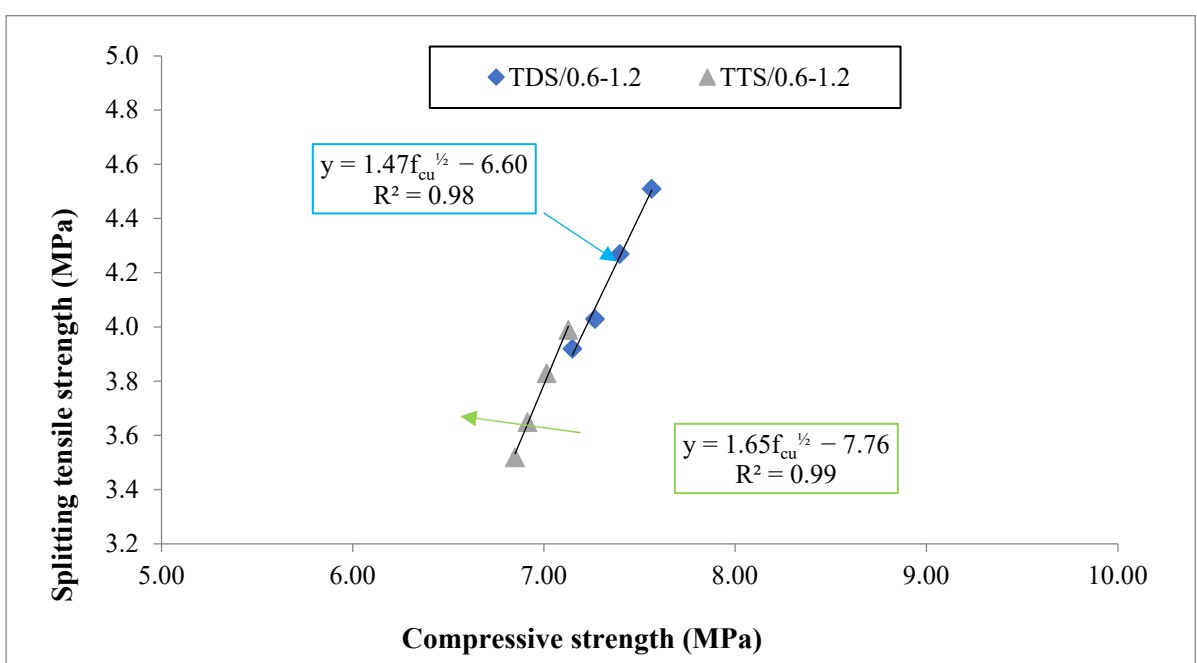

**Figure 12.** Relationship between compressive strength and splitting tensile strength of untreated and treated LWPA concrete.

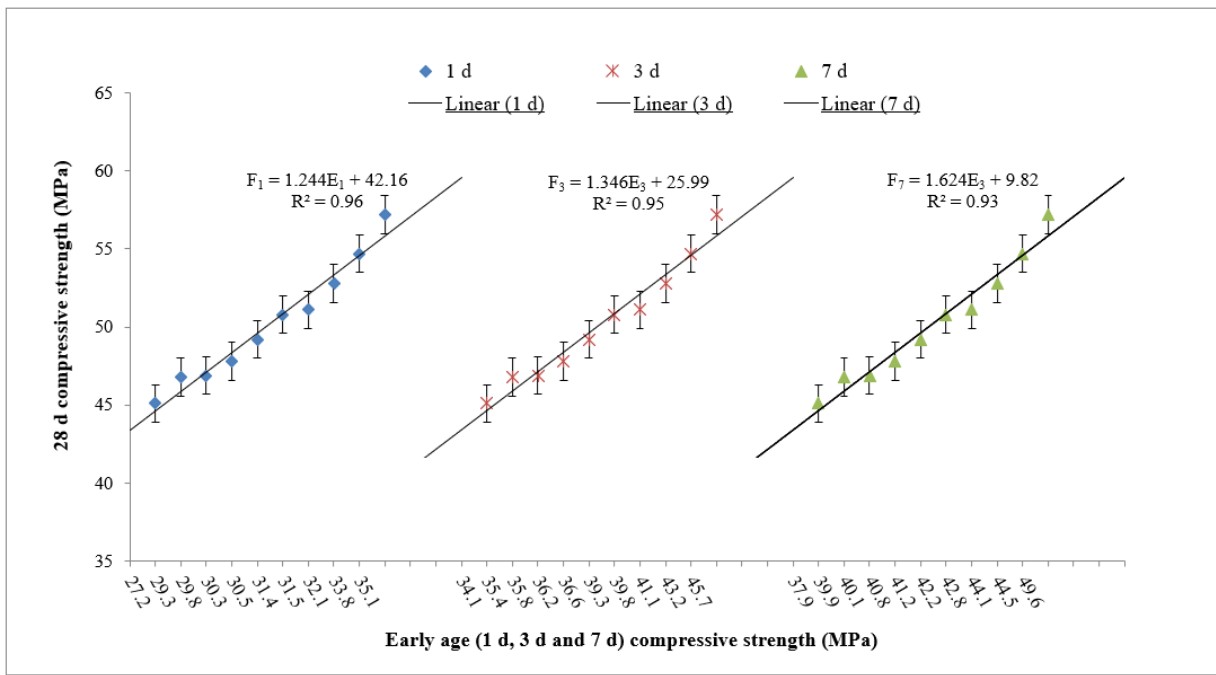

**Figure 13.** Relationship between early stage (1 d, 3 d and 7 d) and 28 d compressive strength of untreated and treated LWPA concrete.

### 3.6. Water Absorption

Untreated and treated LWPA with the pre-soaking method on concrete mixes with the water absorption test are shown in Figure 14. It can be noted that the smallest value of water absorption was about 4.0% for the TTS/0.6 specimen. However, the highest water absorption value was achieved at about 7.7%, specifically for the DS/0 specimen. The water absorption for dura shell concrete, untreated and treated with the pre-soaking method, was higher as compared to tenera shell cube concrete. The dura shell of the oil palm fruit is thicker compared to the tenera shell, which has a thinner and more compact structure.



This phenomenon can be attributed to the thicker dura shell as compared to the tenera shell, which is able to absorb more water. The pre-soaking treatment method with lower w/c ratios may reduce the water absorption process. According to Neville [43], a water absorption rate below 10% is generally considered good and indicates that the concrete is less porous and more resistant to moisture damage. Babu [44] also reported that the addition of expanded polystyrene aggregate achieved a water absorption measurement falling within the range of 3–6%.

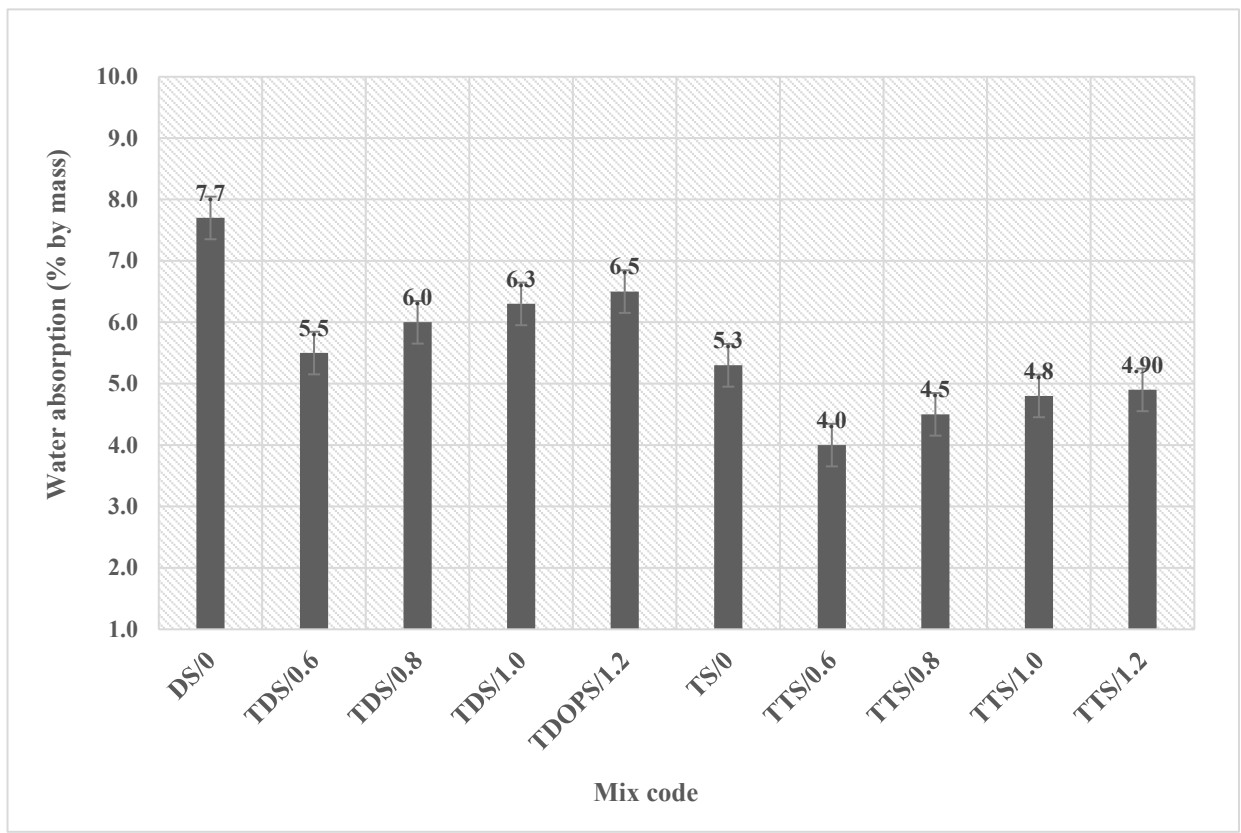

**Figure 14.** Water absorption of untreated and treated LWPA concrete.

### 3.7. Elastic Modulus

The impact of pre-soaking treatment with various w/c ratios on the modulus of elasticity for LWPA concrete at 28 days is shown in Figure 15. The modulus of elasticity for TS/0, DS/0, TTS/0.6–1.2 and TDS/0.6–1.2 mixtures fell in the range of 14.5 GPa–19.4 GPa, respectively. The modulus of elasticity for non-treated LWPC concrete was lower when compared to the pre-soaking treatment method of LWPC concrete. It can be noted that the TDS/0.6 modulus of elasticity increased significantly by approximately 34% compared to DS/0. It is true that the quality of surface pre-soaking treatment modification on LWPA can have a significant impact on the properties of concrete, particularly in the case of the TDS/0.6 mix with surface strengthening. According to Mazaheripour et al. [45], the modulus of elasticity values of normal weight concrete (NWC) fell in the range of 14 GPa–41 GPa. The use of an expanded clay aggregate as a lightweight aggregate in concrete has shown to have favourable results in terms of (E) value. It has been reported that the (E) value for lightweight concrete containing expanded clay aggregate generally falls within the range of 10–14 GPa [46].

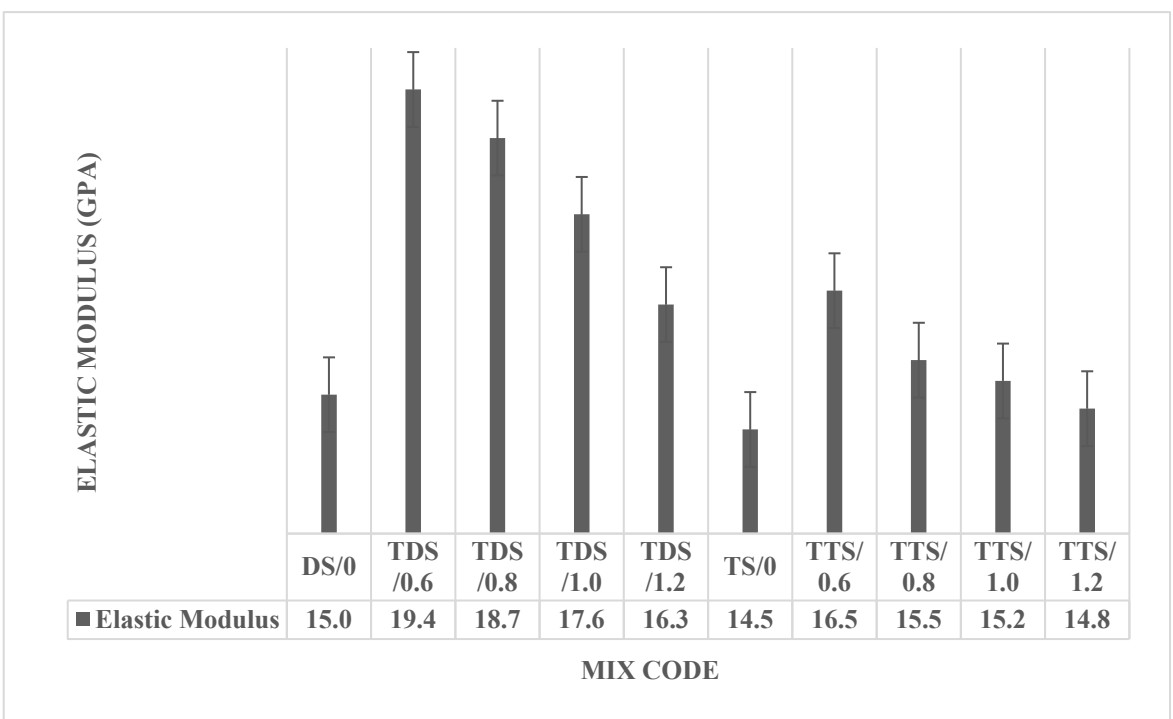

**Figure 15.** Elastic modulus of untreated and treated LWPA concrete.

### 4. Conclusions

In this research, the following conclusions can be arrived at based on the obtained results:

The workability of the LWPA concrete was enhanced when pre-soaking with various w/c ratios was applied. In this context, the statement implies that the TDS/0.6 mix had the highest workability among the mixes tested, as it had a slump value of 155 mm. In addition, incorporating pre-soaking treated LWPA into concrete can slightly increase the density of the concrete.

The impact of pre-soaking treatment on the cube compressive strength were more noticeable at the latter stages. The outcomes of cube compressive strength and split tensile strength of pre-soaking treated LWPA concrete were found to increase significantly when compared to untreated dura and tenera shells.

LWPA concrete exposed to pre-soaking treated with various ratios prove to be in a linear relationship with high correlation coefficients. The water absorption test is an important indicator of the quality of concrete. In the case of lightweight plant-based concrete, the treated and untreated LWPA cube specimens have shown good results in this test, not more than 10% in all concrete specimens.

The inclusion of pre-soaking treatment in LWPA concrete had a positive effect on the modulus of elasticity. The highest ($E$) value was obtained at 19.4 GPa, which increased significantly at about 34% for TDS/0.6 when compared to the control concrete.

**Author Contributions:** Conception, methodology and design of study: M.K.Y., M.C.Y., J.H.B. and F.W.L. Acquisition of data: M.K.Y., M.C.Y., S.K.L. and J.H.L. Analysis and interpretation of data: M.K.Y., M.C.Y., J.H.B., Y.L.L. and K.I.S.A.K. Drafting the manuscript: M.K.Y., J.H.B., M.C.Y. and F.W.L. All authors have read and agreed to the published version of the manuscript.

**Funding:** This publication was supported by IPSR/RMC/UTARSF-SP/2022/003 and IPSR/RMC/UTARRF/2020-C2/Y02 from the Universiti Tunku Abdul Rahman, Malaysia.

**Institutional Review Board Statement:** Not applicable.

**Informed Consent Statement:** Not applicable.

**Data Availability Statement:** The datasets generated and/or analysed during the current study are available from the authors.

**Acknowledgments:** The authors acknowledge Yew See Hing for the supply of the plant-based (dura and tenera) coarse aggregate for this research.

**Conflicts of Interest:** There is no conflict of interest between authors regarding the publication of this paper.

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
