# Peer review of "Effect of Pre-Soaking Treatment Method of Plant-Based Aggregate on the Properties of Lightweight Concrete—Preliminary Study"

_coatings, doi:10.3390/coatings13050864_

Round 1

Reviewer 1 Report

The reviewer thanks the authors and editors for the opportunity to review the manuscript. The article discusses the potential use of plant-based aggregates for the production of lightweight concrete.

General comments:

1) I have doubts about the fit between the content of the article and the scope of the journal. I think that the article does not fit in the Coatings journal. The authors should consider publishing in Materials or Applied Sciences journals.

2) The quality and clarity of all figures should be improved.

3) The article includes general information that confirms the validity of using OPS for lightweight concrete. In the other hand, there is a lack of detailed analyses in this aspect. The authors should conduct more extensive analyses and present them. They should also detail the research methodology.

Specific comments:

Title: I suggest clarifying the title and changing it to: Effect of pre-soaking treatment method of plant-based aggregate on the properties of lightweight concrete - preliminary study

Section 1.2: What are the 1, 3, 7, 28 and 56 day periods derived from? Are they standardised in any way?
The authors mention BS EN 12390-3, BS EN 12390-6, BS EN 12390-13. What do these standards relate to? The authors should describe the test methodology briefly.

Section 2.1.1: What is the slump test method?

Section 2.1.2: The authors indicate undefined categories of lightweight concrete. What is the source of this classification? What are the categories of lightweight concrete?

Section 3.3.1.: It is difficult to indicate a correlation for 4 points. It is too small a number of points.

Section 3: Modest conclusions. The results of the study should be highlighted more widely.

Author Response

Specific comments:

Title: I suggest clarifying the title and changing it to: Effect of pre-soaking treatment method of plant-based aggregate on the properties of lightweight concrete - preliminary study

Answer: The title has been revised accordingly.

Section 1.2: What are the 1, 3, 7, 28 and 56 day periods derived from? Are they standardised in any way?
The authors mention BS EN 12390-3, BS EN 12390-6, BS EN 12390-13. What do these standards relate to? The authors should describe the test methodology briefly.

Answer: All the concrete specimens are kept in a water tank for a duration of 1st, 3rd, 7th, 28th and 56th days for water curing before testing according to different tests as stated. This is the basic requirements/standards for all the concrete samples should put under water tank for water curing before testing. Different tests are performed according to compressive strength: BS EN 12390-3 [36], splitting tensile strength: BS EN 12390-6 [37], modulus of elasticity: BS EN 12390-13 [38] respectively by using compression machine with the capacity up to 3000 kN. All these tests are using compression machine to obtain the results according to standard code as stated in the References.

Section 2.1.1: What is the slump test method?

Answer: The slump test is a method used to determine the consistency of fresh concrete before it sets.

Section 2.1.2: The authors indicate undefined categories of lightweight concrete. What is the source of this classification? What are the categories of lightweight concrete?

Answer: Regarding the densities you mentioned, "demoulded" is not a standard category of density for lightweight concrete. "Air-dry" and "oven-dry" densities are used to describe the drying process of the concrete rather than its overall density. Therefore, all these categories of lightweight concrete play an important role during mixing and casting on-site as references.

Section 3.3.1.: It is difficult to indicate a correlation for 4 points. It is too small a number of points.

Answer: The correlation between the early age strength at 1, 3, and 7 days, and the 28-day strength for untreated and treated lightweight plant-based concrete is significant, as shown in Fig. 13.

Section 3: Modest conclusions. The results of the study should be highlighted more widely.

Answer: Thanks for the suggestion. The conclusions have been revised as highlighted with yellow color.

Reviewer 2 Report

The problems associated with the curing of concrete with plant- based aggregates and OPC are well known. The authors are recommended to add to Section 1 the following information:

-      data on curing parameters of the plant-based aggregates after pre-soaking treatment (temperature, time interval);

-      data on curing of the concrete samples after molding or make references to the standards used in the study.

Author Response

-      data on curing parameters of the plant-based aggregates after pre-soaking treatment (temperature, time interval);

Answer: It's important to note that different types of plant-based aggregates may require different soaking times and temperatures. Therefore, plan-based aggregates after pre-soaking treatment with heating temperature up to 75 °C and time interval 0.5 hour will be conducted.

-      data on curing of the concrete samples after molding or make references to the standards used in the study.

       Answer: The fresh concrete was put in steel molds, which have been properly lubricated before placing. The 100 x 100 x 100 mm cube, 100 mm x 200 mm cylinder and 150 mm x 300 mm cylindrical specimens were used to find the compressive strength, splitting tensile strength and young modulus elasticity, and all the concrete specimens are kept in a water tank for a duration of 1st, 3rd, 7th, 28th and 56th days for water curing. Fresh properties of concrete were examined immediately after mixing. Besides, different tests are performed according to compressive strength: BS EN 12390-3 [36], splitting tensile strength: BS EN 12390-6 [37], modulus of elasticity: BS EN 12390-13 [38] respectively by using compression machine with the capacity up to 3000 kN.

Reviewer 3 Report

Lightweight concrete is a civil engineering topic with societal and economic significance. The manuscript shows substantial improvements in workability and mechanical properties of pre-soaking plant-based light weight aggregate (LWPA) over untreated LWPA concrete. Compressive strength, split tensile strength, modulus of elasticity and interfacial bonding are studied with insights worth disseminating in the community. However, the manuscript needs enhancing in the following aspects.

(1) In the Introduction part, the purpose of each paragraph should be more stressed and the novelty of this work should go deeper, i.e. what is new in this work while the other literatures have not conducted. In addition, reviews on pre-soaking treatment techniques are very important but lacked in the manuscript, and the authors should avoid such a statement as “no information is available regarding the pre-soaking treatment techniques on plant-based aggregates”. At the end of this part, the authors should at least introduce what they plan to do in the work.

(2) An experimental flowchart is required at the beginning of Materials and Methods to provide more multi-dimensional and clear information. The figures in the manuscript need enhancing the resolutions.

(3) Figs. 9 and 10 show the micro/meso-scale structures of the materials, but they are only 2D slices. More insights will be revealed if the authors use Computed Tomography scanning, which is non-destructive to elucidate 3D fracture mechanisms at micro/meso-scale, e.g., Int J Solids Struct, 2015, 67: 340-352, 10.1016/j.cemconcomp.2021.104347. The authors should at least comment on this to make the manuscript more comprehensive.

(4) The language should be polished further. There are some places to be improved, for example, L196 “Martirena et al. [41] also reported the similar trend with positive effect of surface coating treated on recycled aggregate” can be changed to “Martirena et al. [41] also reported similar positive effects provided by surface coating for recycled aggregates”.

(5) Page 13: The Concluding part should be arranged in a more logical manner, because the individual paragraphs in the manuscript look scattered and unrelated to each other.

The language should be polished further. There are some places to be improved, for example, L196 “Martirena et al. [41] also reported the similar trend with positive effect of surface coating treated on recycled aggregate” can be changed to “Martirena et al. [41] also reported similar positive effects provided by surface coating for recycled aggregates”.

Author Response

Lightweight concrete is a civil engineering topic with societal and economic significance. The manuscript shows substantial improvements in workability and mechanical properties of pre-soaking plant-based light weight aggregate (LWPA) over untreated LWPA concrete. Compressive strength, split tensile strength, modulus of elasticity and interfacial bonding are studied with insights worth disseminating in the community. However, the manuscript needs enhancing in the following aspects.

(1) In the Introduction part, the purpose of each paragraph should be more stressed and the novelty of this work should go deeper, i.e. what is new in this work while the other literatures have not conducted. In addition, reviews on pre-soaking treatment techniques are very important but lacked in the manuscript, and the authors should avoid such a statement as “no information is available regarding the pre-soaking treatment techniques on plant-based aggregates”. At the end of this part, the authors should at least introduce what they plan to do in the work.

Answer: Therefore, implementation of innovative techniques on plant-based aggregate with the consideration of effective methods to mitigate environmental issue should be strongly recommended. Thus, the influences of pre-soaking techniques with various w/c ratios formulation (0.6, 0.8, 1.0 and 1.2) on dura and tenera plant-based aggregates in terms of mechanical and durability properties will be investigated.

(2) An experimental flowchart is required at the beginning of Materials and Methods to provide more multi-dimensional and clear information. The figures in the manuscript need enhancing the resolutions.

Answer: Thanks for the suggestion. The flowchart has been revised and move to section 1. Materials and Methods. In addition, the mixing method of untreated and treated LWPA concrete was performed, as shown in Figure 4.

(3) Figs. 9 and 10 show the micro/meso-scale structures of the materials, but they are only 2D slices. More insights will be revealed if the authors use Computed Tomography scanning, which is non-destructive to elucidate 3D fracture mechanisms at micro/meso-scale, e.g., Int J Solids Struct, 2015, 67: 340-352, 10.1016/j.cemconcomp.2021.104347. The authors should at least comment on this to make the manuscript more comprehensive.

Answer: Thanks for the suggestion. It would be good for us to propose the Computed Tomography scanning equipment to the University in the next year budget. However, the SEM images of shell aggregate before and after pre-soaking treated, as shown in Fig. 10. It should be noted that, the contributed of the ‘soaking’ effect that helped to improve the ITZ of lightweight concrete during the strengthening process. In addition, the outcome of the cube concrete especially Fig. 8. Treated (a) and untreated (b) crack surfaces of OPS aggregate for cube compressive strength test has been shown to proof it.

(4) The language should be polished further. There are some places to be improved, for example, L196 “Martirena et al. [41] also reported the similar trend with positive effect of surface coating treated on recycled aggregate” can be changed to “Martirena et al. [41] also reported similar positive effects provided by surface coating for recycled aggregates”.

Answer: Thanks for the comment. It has been revised accordingly. Martirena et al. [41] also reported similar positive effect provided by surface coating for recycled aggregates.

(5) Page 13: The Concluding part should be arranged in a more logical manner, because the individual paragraphs in the manuscript look scattered and unrelated to each other.

Thanks for the suggestion. The conclusions have been revised as follows:

Answer:

1.  Conclusion

In this research, the following conclusions can be arrived at based on the obtained results.:

  • The workability of the LWPA concrete enhanced when pre- soaking with various w/c ratios applied. In this context, the statement implies that the TDS/0.6 mix had the highest workability among the mixes tested, as it had a slump value of 155 mm. Besides, incorporating pre-soaking treated LWPA into concrete can slightly increase the density of the concrete.

  • The impacts of pre-soaking treated on the cube compressive strength were more noticeable at the latter stages. The outcomes of cube compressive strength and split tensile strength of pre-soaking treated LWPA concrete was found increased significantly when compared to untreated dura and tenera shells.

  • LWPA concrete exposed to pre-soaking treated with various ratios proven a linear relationship with high correlation coefficients. The water absorption test is an important indicator of the quality of concrete. In the case of lightweight plant-based concrete, the treated and untreated LWPA cube specimens have shown good results in this test, showing all concrete specimens not more than 10%.

  • The inclusion of pre-soaking treated in LWPA concrete implied a positive effect on the modulus of elasticity. The highest (E) value was obtained at 19.4 GPa, which increased significantly at about 34% for TDS/0.6 when compared to control concrete.

Comments on the Quality of English Language

The language should be polished further. There are some places to be improved, for example, L196 “Martirena et al. [41] also reported the similar trend with positive effect of surface coating treated on recycled aggregate” can be changed to “Martirena et al. [41] also reported similar positive effects provided by surface coating for recycled aggregates”.

Thanks for the suggestion. All the comments have been revised accordingly as highlighted in yellow color in the full manuscript.

Round 2

Reviewer 1 Report

The reviewer again would like to thank the authors and the editors for the opportunity to review the manuscript. The manuscript shows an improvement in the text and an increase of its quality. I recommend the manuscript for publication.

Reviewer 3 Report

The manuscript can be accepted in the present form.

Minor editing is required in the proofing.